# Nutritional and Functional Advantages of the Use of Fermented Black Chickpea Flour for Semolina-Pasta Fortification

**DOI:** 10.3390/foods10010182

**Published:** 2021-01-18

**Authors:** Ilaria De Pasquale, Michela Verni, Vito Verardo, Ana María Gómez-Caravaca, Carlo Giuseppe Rizzello

**Affiliations:** 1Department of Soil, Plant and Food Science, University of Bari Aldo Moro, 70121 Bari, Italy; ilaria.depasquale@uniba.it; 2Department of Nutrition and Food Science, Campus Universitario de Cartuja, University of Granada, E-18071 Granada, Spain; vitoverardo@ugr.es; 3Institute of Nutrition and Food Technology ‘José Mataix’, Biomedical Research Centre, University of Granada, Avenida del Conocimiento s/n, E-18071 Granada, Spain; 4Department of Analytical Chemistry, University of Granada, Avda Fuentenueva s/n, E-18071 Granada, Spain; anagomez@ugr.es; 5Department of Environmental Biology, Sapienza University of Rome, 00185 Roma, Italy; carlogiuseppe.rizzello@uniroma1.it

**Keywords:** black chickpea, pasta, lactic acid bacteria, fermentation, phenolic compounds

## Abstract

Pasta represents a dominant portion of the diet worldwide and its functionalization with high nutritional value ingredients, such as legumes, is the most ideal solution to shape consumers behavior towards healthier food choices. Aiming at improving the nutritional quality of semolina pasta, semi-liquid dough of a Mediterranean black chickpea flour, fermented with *Lactiplantibacillus plantarum* T0A10, was used at a substitution level of 15% to manufacture fortified pasta. Fermentation with the selected starter enabled the release of 20% of bound phenolic compounds, and the conversion of free compounds into more active forms (dihydrocaffeic and phloretic acid) in the dough. Fermented dough also had higher resistant starch (up to 60% compared to the control) and total free amino acids (almost 3 g/kg) contents, whereas antinutritional factors (raffinose, condensed tannins, trypsin inhibitors and saponins) significantly decreased. The impact of black chickpea addition on pasta nutritional, technological and sensory features, was also assessed. Compared to traditional (semolina) pasta, fortified pasta had lower starch hydrolysis rate (*ca*. 18%) and higher in vitro protein digestibility (up to 38%). Moreover, fortified cooked pasta, showing scavenging activity against DPPH and ABTS radicals and intense inhibition of linoleic acid peroxidation, was appreciated for its peculiar organoleptic profile. Therefore, fermentation technology appears to be a promising tool to enhance the quality of pasta and promote the use of local chickpea cultivars while preventing their genetic erosion.

## 1. Introduction

Globally, pasta represents one of the most popular and consumed food, and its production is associated to a relatively simple and inexpensive technology [1]. The last report of the International Pasta Organization (IPO) indicated that 14.5 million tons of pasta were produced worldwide in 2018 [2], with revenues of over US$99 billion and an expected growth rate of 2.7% annually till 2025 [3]. Pasta is an important source of carbohydrates (mainly starch), it contains proteins, variable amount of fibers and a very low concentration of fat [1]. Pasta is regularly eaten in amounts (average yearly world consumption of 7.2 kg per capita in 2020 [3]) representing a dominant portion of the diet worldwide; therefore, the growing awareness of the need to reduce sugars and fat while increasing proteins and fibers intakes, makes it the most ideal food to be fortified, shaping consumers behavior towards healthier choices.

Pasta production technology easily allows its functionalization, since fortification with different active ingredients such as fibers, polyphenols, minerals or proteins [1] is possible. Numerous attempts to increase the nutritional value of the pasta by adding different cereals (e.g., barley and pigmented cereals) [4,5], legume and pseudocereal flours [6,7,8,9,10] and ingredients from various origins, such as oregano and carrot leaf meal [11], fish (Tilapia) [12], carob [13] and maize bran, grape marc and brewers spent grain flour [14] were already reported.

The use of flours different from wheat is currently considered by the food industry as a successful strategy to meet the demand of healthy and alternative products from an increasing niche of consumers, also due to their positive image and association with natural and traditional ethnic foods [15], and the positive impact on local economy and environment. Legumes have very different chemical composition and technological properties compared to wheat. They are excellent sources of proteins with high biological value, providing many essential amino acids, they contain carbohydrates and dietary fibers, and supply relevant levels of vitamins, minerals, oligosaccharides and phenolic compounds [16].

However, legumes contain heterogeneous and species-dependent antinutritional factors (such as α-galactosides of sucrose, phytic acid, condensed tannins, saponins, trypsin inhibitors, etc.), which can interfere with protein or mineral absorption or sometimes cause pathologic conditions, making necessary their reduction. Several physical treatments (e.g., microwaving, extrusion, steaming, boiling, roasting and infrared) have been proposed [17] to decrease them. Nevertheless, biological methods such as germination, enzyme treatments and especially fermentation by selected lactic acid bacteria (LAB) were reported as more efficient [17,18,19,20,21,22]. Tailored fermentation is the most suitable option not only to improve rheology and sensory attributes, while limiting or eliminating antinutritional factors (ANFs), but also to fully exploit the nutritional and functional potential of vegetable matrices and cereal-based foods with them produced (for a review see Gobbetti et al. [23]).

Among legumes, chickpea (*Cicer arietinum* L.) is the third most produced grain legume worldwide, after common bean and pea. Until 2018, its annual global production was about 17.2 million tons [3]. Chickpea cultivars are grouped in two mains groups: kabuli (lighter color, large seeds and with a smoother coat) and desi (small, colored seeds and a rough coat). In addition, a black-colored chickpea cultivar from Southern Italy (“Apulian black chickpea”) was recently characterized for its peculiar phenotypic and genetic features [24] compared to desi and kabuli types. This cultivar is characterized by small, black, wrinkled, irregular and hooked shape seeds. Compared to desi-type varieties, the black chickpea seeds are bigger and darker [25]. Besides the high protein content, they are a very good source of iron, fibers (three times the amount present in a common chickpea), polyunsaturated fatty acid and bioactive compounds, such as anthocyanins and carotenoids [26].

Based on these considerations, aiming at improving the nutritional quality of semolina pasta, fermented black chickpea flour was used as additional ingredients to manufacture fortified pasta. First, the effects of fermentation with selected LAB on the main nutritional features of black chickpea flour were studied, focusing on antinutritional compounds. Then, the antioxidant activity and the phenolic profile, including identification and quantification of free and bound phenolic compounds, were investigated. Moreover, the effects of the fermented black chickpea fortification on the technological, nutritional, functional, and sensory properties of the derived fortified pasta, were investigated in comparison to conventional semolina pasta and an unfermented fortified control.

## 2. Materials and Methods

### 2.1. Raw Materials and Microorganisms

Black chickpea flour was purchased at a local organic market and was characterized by the following proximal composition: moisture, 9.0%; protein, 18.2% of dry matter (d.m.); fat, 3.7% (d.m.); total carbohydrates, 57.4% (d.m.); total dietary fibers, 17.7% (d.m.) and ash, 3% (d.m.).

*Lactiplantibacillus plantarum* T0A10, formerly classified as *Lactobacillus plantarum* and previously isolated from quinoa flour [27], belonging to the Culture Collection of the Department of Soil, Plant, and Food Science (University of Bari, IT) was used as starter for black chickpea flour fermentation. The strain was routinely propagated at 30 °C in MRS broth (Oxoid, Basingstoke, Hampshire, England).

### 2.2. Fermentation

Black chickpea flour was subjected to spontaneous fermentation (sBC), corresponding to incubation at 30 °C for 24 h without microbial starter addition, or it was inoculated and fermented with the selected LAB (labBC).

Doughs (500 g) were prepared by mixing black chickpea flour with tap water. Dough yield (DY, dough weight × 100/flour weight) was 285, corresponding to 35 and 65 g/100 g of flour and water, respectively. Prior to being inoculated, *L. plantarum* T0A10 was cultivated until the late exponential phase of growth was reached (approximately 12 h). The growth of the strain was estimated by measuring the optical density (OD) at 620 nm using a spectrophotometer model Ultrospec 3000 (Pharmacia Biotech, Sweden), with cuvettes with a 1-cm light path). Microbial kinetics of growth, determined and modeled in agreement with the Gompertz equation, were the following: A (cell density variation between inoculation and the stationary phase), 8.26 ± 0.31; μmax (maximum growth rate expressed as OD_620_ units/h), 1.49 ± 0.08 and λ (length of the lag phase measured in hours), 0.87 ± 0.04. Cells were harvested by centrifugation (10,000× *g*, 10 min, 4 °C), washed twice in 50 mmol/L sterile potassium phosphate buffer (pH 7.0) and resuspended in tap water used for dough preparation (final cell density in the dough was approximately log 7.0 log_10_ cfu/g). Fermentation was carried out at 30 °C for 24 h. A not-incubated and not-inoculated dough (BC) was used as a control. All the doughs were prepared in triplicate. 

### 2.3. Doughs Characterization

#### 2.3.1. Microbiological Analysis

Presumptive LAB were enumerated using MRS agar medium (Oxoid, Basingstoke, Hampshire, United Kingdom) supplemented with cycloheximide (0.1 g/L). Plates were incubated at 30 °C for 48 h, under anaerobiosis (AnaeroGen and AnaeroJar, Oxoid). Cell densities of yeasts and molds were estimated on yeast peptone dextrose agar medium (YPDA, Sigma-Merck, Darmstadt, Germany) supplemented with chloramphenicol (0.1 g/L), through pour and spread plate enumeration, respectively, and incubated at 30 °C for 72 h. The attribution (yeast/mold) was performed by visual analysis of colony morphology. Total Enterobacteriaceae were determined on violet red bile glucose agar (VRBGA, Oxoid) at 37 °C for 24 h. 

#### 2.3.2. Biochemical Characterization 

The pH of doughs was determined by a pH meter (Model 507, Crison, Milan, Italy) with a food penetration probe and total titratable acidity (TTA) was determined according to AACC method 02–31.01 [28]. 

Water/salt-soluble extracts (WSE) from doughs were prepared according to the method originally described by Osborne and modified by Weiss et al. [29]. Fifteen grams of sample were resuspended in 60 mL of 50 mM Tris–HCl (pH 8.8), held at 4 °C for 1 h, under stirring condition (150 rpm) and centrifuged at 20,000× *g* for 20 min. The supernatant was used for analyses. WSE were used to determine total free amino acids (TFAA) and organic (lactic and acetic) acids concentration. TFAA were analyzed by a Biochrom 30^+^ series Automatic Amino Acid Analyzer (Biochrom Ltd., Cambridge Science Park, UK), equipped with a Li-cation-exchange column (4.6 mm × 200 mm internal diameter), using lithium citrate buffer eluents following the elution conditions recommended by the manufacturer. A mixture of amino acids at known concentrations (Sigma Chemical Co., Milan, Italy) was added with tryptophan, ornithine and γ-aminobutyric acid (GABA) and used as a standard. Proteins and peptides in the samples were precipitated by addition of 5% (*v/v*) cold solid sulfosalicylic acid, holding the samples at 4 °C for 1 h and centrifuging them at 15,000× *g* for 15 min. Organic acids were determined by high performance liquid chromatography (HPLC), using an ÄKTA Purifier system (GE Healthcare, Buckinghamshire, UK) equipped with an Aminex HPX-87H column (ion exclusion, Biorad, Richmond, CA, USA), and an UV detector operating at 210 nm. Elution was carried out at 60 °C, with a flow rate of 0.6 mL/min, using H_2_SO_4_ 10 mM as the mobile phase. The fermentation quotient (FQ) was determined as the molar ratio between lactic and acetic acids. Resistant starch of doughs, prior and after the fermentation, was determined according to the AACC approved methods 32–40.01 [28].

#### 2.3.3. In Vitro Protein Digestibility

The in vitro protein digestibility (IVPD) was determined on doughs by the method proposed by Akeson and Stahmann [30] with some modifications [27]. Samples were subjected to a sequential enzyme treatment mimicking the in vivo digestion in the gastro-intestinal tract and IVPD was expressed as the percentage of the total protein, which was solubilized after enzyme hydrolysis. The concentration of protein of digested and non-digested fractions was determined by the Bradford method [31].

#### 2.3.4. Antioxidant Activity

The antioxidant activity was measured by three different methods on WSE and methanolic extract (ME). Five grams of each sample were mixed with 50 mL of 80% methanol to get ME. The mixture was purged with nitrogen stream for 30 min, under stirring condition, and centrifuged at 4600 × *g* for 20 min. MEs were transferred into test tubes, purged with a nitrogen stream and stored at ca. 4 °C before analysis. The 2,2-diphenyl-1-picrylhydrazyl (DPPH) radical scavenging activity was determined on ME as described by Yu et al. [32]. The scavenging activity was expressed as follows: DPPH scavenging activity (%) = ((blank absorbance − sample absorbance)/blank absorbance) × 100. The value of absorbance was compared with 75 ppm butylated hydroxytoluene (BHT), which was used as the antioxidant reference.

The radical cation (2,2′-azino-di-[3-ethylbenzthiazoline sulphonate]) (ABTS+) scavenging capacity of MEs was measured using the Antioxidant Assay Kit CSO790 (Sigma Chemical Co.), following the manufacturer’s instruction. Trolox (6-hydroxy2,4,7,8-tetramethylchroman-2-carboxylic acid) was used as the antioxidant standard. The scavenging activity was expressed as the Trolox equivalent. A negative control (without antioxidants) also was considered.

Lipid peroxidation inhibitory activity of WSE was also measured. After freeze-drying, 1.0 mg of each sample was suspended in 1.0 mL of 0.1 M phosphate buffer (pH 7.0) and added to 1 mL of linoleic acid (50 mM), which was previously dissolved in ethanol (99.5%). Incubation in a glass test tube, tightly sealed with a silicon rubber cap, was allowed at 60 °C in the dark for 8 days. The degree of oxidation was determined by measuring the values of ferric thiocyanate according to the method described by Mitsuta et al. [33]. One hundred microliters of the sample were mixed with 4.7 mL of 75% (vol/vol) ethanol, 0.1 mL of 30% (*w/v*) ammonium thiocyanate and 0.1 mL of 0.02 M ferrous chloride, which had been dissolved in 1 M HCl. After 3 min, the degree of color development, representing the oxidation of linoleic acid, was measured spectrophotometrically at 500 nm. BHT (1 mg/mL) was also assayed as antioxidant references. A reference sample (without the addition of antioxidants) was included in the assay as a negative control.

#### 2.3.5. Extraction, Identification and Quantification of Free and Bound Phenolic Compounds

Free and bound phenolics were extracted as described by Verardo et al. [34]. Briefly, 4 g of dried sample were extracted twice in an ultrasonic bath with ethanol/water (4:1 *v/v*) for 10 min. The supernatants were collected, evaporated at 40 °C in a rotary evaporator and reconstituted with 2 mL of methanol/water (1:1 *v/v*). The extracts were stored at −18 °C until use. 

Residues of free phenolics extraction were digested with 300 mL of 2 M NaOH at room temperature overnight by shaking under nitrogen gas. At the end of the incubation, the mixtures were acidified (pH 2–3) with hydrochloric acid and extracted with diethyl ether/ethyl acetate (1:1 *v/v*). The organic fractions were pooled and evaporated to dryness at 40 °C in a rotary evaporator and bound phenolic compounds were reconstituted in 2 mL of methanol/water (1:1 *v/v*), as well.

The identification and quantification of free and bound polyphenols were carried out with the use of an ACQUITY Ultra Performance LC system equipped with a photodiode array detector with a binary solvent manager (Waters Corporation, Milford, MA, USA) series with a mass detector Q/TOF micro mass spectrometer (Waters) equipped with an electrospray ionization (ESI) source operating in the negative mode as described in Verni et al. [35]. The compounds were monitored at 280 nm. Integration and data elaboration were performed using MassLynx 4.1 software (Waters Corporation, Milford, MA, USA). For the quantification of phenolic compounds, solutions of ferulic acid, chlorogenic acid, catechin and quercetin in methanol were prepared and used as a standard. 

#### 2.3.6. Antinutritional Factors

Raffinose was measured using Raffinose/D-Galactose Assay Kit K-RAFGA (Megazyme International Ireland Limited, Bray, Ireland), following the manufacturer’s instructions. 

Condensed tannins were determined using the acid butanol assay, as described by Hagerman [36]. Trypsin inhibitors were determined as described by Alonso et al. [37], using α-N-benzoyl-DL-arginine-*p*-nitroanilidehydrochloride (BA*p*NA) as the substrate for trypsin. Trypsin inhibitor activity (TIA), expressed as trypsin inhibitor units/mg sample, was calculated from the absorbance read at 410 nm against a reagent blank. One trypsin unit was determined as the increase by the 0.01 absorbance unit at 410 nm of the reaction mixture. 

Total saponins in flour and sourdough were determined as reported by Lai et al. [38] with some modifications. Briefly, the freeze-dried samples (0.5g) were mixed with 10mL of petroleum ether by shaking for 4h. The residues (20mg) were then extracted with 5mL of 80% (*v/v*) aqueous methanol with shaking for 4h. The extracts were kept at 4 °C in the dark until they were subjected to analysis. Total saponin content (TSC) was determined using the spectrophotometric method [38]. All data were expressed on a dry weight basis.

### 2.4. Pasta Making

Experimental pasta samples were produced by using a pilot plant La Parmigiana SG30 (Fidenza, Italy). Table 1 summarized the pasta formulations. All the doughs were made with a final DY of 130, corresponding to the mixture of 23% water and 77% of semolina/black chickpea flours.

The fermented black chickpea dough labBC was obtained as described before and used as an ingredient for pasta making (labBC-P). It was used at 15% (*w/w*) of the pasta formulation, corresponding to 6.8% semolina replacement on dry weight (5.3% of the final dough weight). An experimental pasta including the same amount of the unfermented black chickpea dough BC was produced (BC-P). Wheat semolina (*Triticum durum*) flour was purchased at the local organic market. The proximate composition was moisture, 10.2%; protein, 13.5% of dry matter (d.m.) g; fat, 2.18% of d.m.; total carbohydrates, 79.2% of d.m.; total dietary fibers, 4.2% of d.m. and ash, 0.8% of d.m. A reference pasta was made only using wheat semolina (WP).

Ingredients were mixed in three steps (1 min mixing and 6 min hydration). Then, the final dough was mixed for 30 s and extruded at 45–50 °C, through an n.76 bronze die (150 mm diameter). The extruded material was cut with a rotating knife for short pasta shapes to obtain grooved “macaroni”. After extrusion, pasta was arranged on frames (1.5 kg for frame) and treated according to the cycle at low temperature (up to 55 °C, for ca. 8 h), as described in (Appendix A).

### 2.5. Pasta Characterization

#### 2.5.1. Hydration Test, Cooking Time, Cooking Loss and Water Absorption

Hydration in water was determined at 25 °C (ratio pasta: water of 1:20), after 180 min of incubation [39]. The optimal cooking time (OCT) was identified as the boiling time necessary to the disappearance of the white pasta core [40]. Cooking loss, corresponding to the amount of solids recovered from the cooking water, was expressed as grams of matter loss/100 g of pasta [41]. The water absorption during cooking was determined with the following formula: ((W_1_ − W_0_) × 100/W_0_), in which W_1_ and W_0_ corresponded to the pasta weight after and before cooking, respectively. 

#### 2.5.2. Biochemical Characterization and Antioxidant Activity

Moisture, protein (total nitrogen × 5.7), fat, total dietary fiber and ash were determined according to Approved Methods 44-15A, 46-11A, 30–10.01, 32–05.01 and 08–01.01 of the American Association of Cereal Chemists [28]. Available carbohydrates were calculated as the difference (100 − (proteins + fat + ash + total dietary fiber)). Proteins, fat, carbohydrates, total dietary fiber and ash were expressed as % of d.m.

pH, TFAA, total phenols and free and bound phenolic compounds of pasta samples were determined as reported above. The radical scavenging activity towards DPPH and ABTS was determined on ME whereas lipid peroxidation inhibitory activity was measured on WSE as reported in Section 2.3.4. All antioxidant activities were performed on pasta samples cooked until the OCT. 

#### 2.5.3. Protein and Starch Digestibility

IVPD and starch hydrolysis were determined on pasta samples at the OCT. IVPD was determined as described in Section 2.3.3, while the analysis of starch hydrolysis was carried out with a procedure mimicking the in vivo digestion of starch previously proposed by De Angelis et al. [42]. The degree of starch digestion was expressed as a percentage of potentially available starch hydrolyzed at different times (30, 60, 90, 120 and 180 min). Wheat flour bread was used as control to estimate the hydrolysis index (HI = 100). The predicted glycemic index (pGI) was calculated using the equation: pGI = 0.549 × HI + 39.71 [43]. 

#### 2.5.4. Texture and Color Analysis 

Instrumental Texture Profile Analysis (TPA) was performed with a TVT-300XP Texture Analyzer (TexVol Instruments, Viken, Sweden). A cylinder probe 95 mm diameter was used. Before analysis, pasta was cooked (at the OCT), drained of excess water, cooled to room temperature and then placed in a beaker (diameter, 100 mm; height 90 mm). Sample filled half the beaker volume. The analysis was carried out in two compression cycles by using the following parameters: test speed 1 mm/s, 30% deformation of the sample [9]. 

The color analysis was carried out with a Minolta CR-10 camera. Chromaticity coordinates L, a and b were determined, and used to calculate the color difference value, ΔE*_ab_, by using the following equation. ΔL, Δa and Δb are the differences for L, a and b values between sample and a white ceramic plate using as reference (L = 93.4, a = −1.8 and b = 4.4).
(1)$$\Delta E{*}_{\mathrm{ab}}\;=\;\sqrt{{\left(\Delta L\right)}^{2}+{\left(\Delta a\right)}^{2}+{\left(\Delta b\right)}^{2}}$$

#### 2.5.5. Sensory Analysis

The sensory profile of the experimental pasta was analyzed by a trained sensory panel (*n* = 13, aged 21–45 years), following the ethical guidelines approved for the sensory laboratory. A written informed consent was signed by all the participants. Twelve sensory attributes (described in Appendix A) were included in the lexicon [9]. Evaluation was carried out using a line scale from “not at all” (0) to “very” (10). Samples were analyzed after cooking at corresponding OCT and randomized distributed to the panelists, in duplicate. Between the analysis of the pasta samples, water was served to rinse the mouth.

### 2.6. Statistical Analysis

All the chemical and physical analyses were carried out in triplicate for each batch of dough and pasta samples. Data were subjected to one-way ANOVA; paired comparison of treatment means was achieved by Tukey’s procedure at *p* < 0.05, using the statistical software Statistica 12.5 (StatSoft Inc., Tulsa, OK, USA).

## 3. Results

### 3.1. Black Chickpea Fermentation

#### 3.1.1. Microbiological and Biochemical Characterization

Chickpea flour was used to prepare a semiliquid dough having a DY of 285, by adding tap water. Before fermentation, the dough (BC) was characterized by a LAB density of 2.5 ± 0.12 log cfu/g (Table 2). Slightly lower densities were found for yeasts, molds and Enterobacteriaceae. After incubation at 30 °C (sBC), all the microbial groups significantly (*p <* 0.05) increased, especially LAB, that were found at 7.1 ± 0.16 log cfu/g after 24 h of incubation, while yeasts and Enterobacteriaceae increased of 2 log cycles, and molds of 1 log cycle (Table 2). labBC, that was inoculated with the starter, had an initial LAB density of 7.2 ± 0.3 log cfu/g, that reached 9.6 ± 0.11 log cfu/g after incubation, whereas, yeasts, molds and Enterobacteriaceae did not differ significantly (*p* > 0.05) compared to BC. 

As expected, the pH significantly (*p* < 0.05) decreased during incubation, nevertheless the final pH was significantly (*p* < 0.05) lower in labBC compared to sBC. Accordingly, TTA of the labBC was 2 and 5 times higher than those observed for BC and sBC, respectively (Table 2). Indeed, lactic acid concentration in labBC was 60% higher than the spontaneously fermented sample sBC. Nevertheless, for the latter, a significantly (*p* < 0.05) higher concentration of acetic acid was found compared to the inoculated sample (Table 2). The fermentation quotient of labBC was 9.41.

Overall, very high concentration of free amino acids (FAA) was found in chickpea doughs. The unfermented control contained more than 1.85 g/kg, and significant (*p* < 0.05) increases were observed during incubation. The inoculated samples contained the highest amount of total FAA (Table 2). In particular, BC was characterized by very high concentrations (>348 mg/kg) of Glu, Arg and Orn, while Met, Asp, Thr, Ser and Ala were at concentrations lower than 30 mg/kg (Figure 1). Incubation caused the increase of almost all FAA in sBC, with the exception of Ser, Glu, Ile, Tyr and Arg that decreased by 28, 53, 30, 18 and 38%, respectively (Figure 1). Compared to sBC, labBC was characterized by higher (*p* < 0.05) concentrations of all the FAA except for Asp, Tyr, Orn, Lys, Arg and Cys. In labBC, Glu was the FAA at the highest concentration (435 ± 16 mg/kg), followed by GABA (269 ± 9 mg/kg) and Val (262 ± 10 mg/kg) (Figure 1).

Resistant starch also increased during fermentation. Compared to the control, it was 16 and 60% higher in sBC and labBC, respectively (Table 2).

#### 3.1.2. Digestibility and Antinutritional Factors

IVPD of the chickpea control dough was 80%, and significant (*p* < 0.05) increases were found at the end of incubation in both the fermented samples (reaching up to 91% ± 2% in labBC). 

All the ANF decreased during incubation, especially when black chickpea flour was fermented with the selected starter (Table 2). Indeed, in labBC, raffinose decreased up to 65% compared to control, while condensed tannins and saponins were halved. Compared to BC, TIA decreased of 11% and 20% in sBC and labBC, respectively (Table 2).

#### 3.1.3. Antioxidant Activity

Antioxidant activity resulted in any case higher in incubated samples compared to the not fermented control dough. Indeed, the radical scavenging activity, as determined on the DPPH radical using the methanolic extracts of the doughs, significantly (*p* < 0.05) increased by 23 and 64% in sBC and labBC compared to BC (Figure 2A). The radical cation ABTS scavenging activity was similar (*p* > 0.05) for BC and sBC, while an increase of the 25% was found when *L. plantarum* was used as a starter for fermentation (Figure 2B). For all samples, values higher than 80% were found for the inhibition of the lipid peroxidation as determined on linoleic acid after 8 days of incubation, reaching up to 94% in labBC (Figure 2C). Peroxidation inhibition in the presence of BHT, used as a reference, was 82.5% ± 0.5%.

#### 3.1.4. Identification of Free and Bound Phenolic Compounds 

Free and bound phenolic compounds were separated and identified by UPLC-PDA-ESI-QTOF. Appendix A summarizes the information related to the compounds tentatively identified: retention times, experimental and calculated *m/z*, molecular formula, fragments, score and error (ppm). Thirty-four phenolic compounds were identified among free and bound profiles.

In free phenolic profile, compound **1**, showing molecular ion at *m/z* 153 and molecular formula C_7_H_6_O_4_, was identified as protocatechuic acid, previously found in chickpea fractions by Sreerama et al. [44]. Four compounds with molecular ions at *m/z* 315, 431 and 447 (isomers I and II), detected in both free and bound extracts, showing same [M-H]^-^ and fragments of that found by Mekki et al. [45], were identified as dihydroxybenzoic acid hexoside, hydroxybenzoic acid hexoside-pentoside and dihydroxybenzoic acid hexoside-pentoside, respectively. Except for raw chickpea dough, in the free phenolic profile, compound **3** and **9**, showing *m/z* at 181 and 165, were identified as dihydrocaffeic and phloretic acids. In the free phenolic profile, a peak showing molecular ion at *m/z* 186 and molecular formula C_11_H_9_N_1_O_2_, was identified as indole-3-acrylic acid, previously identified in other legume species [46]. Two isomers (compound **8** and **10**), at *m/z* 281, were identified as dihydrophaseic acids, previously found in several chickpea Egyptian cultivars [45]. Compounds **11** and **12**, showing *m/z* 845 and 1259 and providing fragments in MS^2^ signal at *m/z* 431, 137 and 845, 431 and 137, respectively, were tentatively identified as dehydrodimer and trimer of hydroxybenzoic acid hexoside-pentoside. Whereas compound **13**, providing a molecular ion at *m/z* 551, and fragments at *m/z* 413 and 137 in MS^2^ signal, was tentatively identified as hydroxybenzoic acid derivative. Five more compounds (**18**, **19**, **20**, **21** and **22**), showing the same [M-H]^-^ and fragments of that previously identified by Mekki et al. [45], were identified as quercetin, myricetin and kaempferol glycosides. Another compound, providing a molecular ion at *m/z* 757 and fragment at *m/z* 316, with molecular formula C_15_H_8_O_8_, was tentatively identified as myricetin derivative. The last two compounds (**20** and **21**), showing *m/z* at 1081 and 941, respectively, were identified as saponins, previously described in the Fabaceae family, including chickpea [45,47].

In the bound phenolic profile, two compounds providing a molecular ion at *m/z* 169 and 153 were identified as gallic and protocatechuic acids, respectively. Two isomers compounds (**8** and **9**) with a molecular ion at *m/z* 193 were identified as ferulic and isoferulic acid as confirmed by coelution with ferulic acid commercial standard and retention times of a work carried out in a similar condition [48]. All the hydroxycinnamic and hydroxybenzoic acids identified were previously found in chickpea fractions by Sreerama et al. [44]. Peak 5, which provided *m/z* 301 and molecular formula C_15_H_10_O_7_, was identified as the quercetin isomer morin, based on the retention times described in Ali et al. [49]. Compound **7**, **10** and **11**, as for free phenolics, were identified as dihydrophaseic acid, myricetin-O-methyl ether hexoside-deoxyhexoside-pentoside and kaempferol 3-O-lathyroside-7-O-α-L-rhamnopyranoside, respectively. Two more compounds (**12** and **13**), showing *m/z* at 955 and 941, respectively, were identified as soyasaponin Ba and Bd, also previously described in chickpea [45,47].

#### 3.1.5. Quantification of Phenolic Compounds

Free and bound phenolic profiles were characterized by different compounds both in terms of quality and quantity. Glycosides of hydroxybenzoic acid and flavonols were made up respectively for 37% and 35% of the total free phenolic compounds quantified in the raw matrix (Table 3). Oligomers and derivatives of hydroxybenzoic acid were also detected for a total of 3.5 mg/kg d.m., whereas among flavonols glycosides, myricetin-*O*-methyl ether hexoside-deoxyhexoside-pentoside was the most represented, reaching up to 13% of the total free compounds. A similar concentration (3.72 ± 0.19 mg/kg d.m.) was found for protocatechuic acid. Dihydrocaffeic and phloretic acids, which were not detected in BC, were the most representative compounds in sBC and labBC. Spontaneously fermented and inoculated chickpea doughs, compared to BC, had a concentration of total free phenolic compounds **3**- and **9**-fold higher, respectively. Except for dihydroxybenzoic acid hexoside, which was undetected in processed samples, and two quercetin glycosides, which significantly decreased *(p* < 0.05), all the other compounds increased (from 20% to 3 times), especially when *L. plantarum* was used (Table 3).

As for bound phenolics, which had a concentration almost 3 times higher than free phenolics, phenolic acids were the most representative, reaching up to 92% of the total bound compounds in BC, of which protocatechuic and isoferulic acids were the most abundant. Hydroxybenzoic acids glycosides and glycosides of myricetin and kaempferol were detected in a small amount, reaching up to 5 mg/kg d.m. An opposite trend, compared to free phenolics, was observed for processed samples, where a decrease from 20 to 35% found, evenly distributed among compounds (Table 3).

### 3.2. Fortified Pasta 

#### 3.2.1. Technological Characterization 

The kinetics of water uptake at 25 °C showed no statistical differences (*p* > 0.05) among pasta samples up to 30 min of incubation. Then, compared to WP (47 ± 2 g/100), the hydration of BC-P and labBC-P increased reaching 63 ± 2 and 90 ± 4 g/100 g in BC-P and labBC, respectively, at 180 min. The experimental OCT of semolina control pasta WP was 9.7 ± 0.2 min. Significantly (*p* < 0.05) lower values were observed for fortified samples (29% and 35% respectively for BC-P and labBC-P). Compared to WP, also water absorption significantly decreased for both fortified pasta samples (Table 4). On the contrary, cooking loss was 23% higher in BC-P and labBC-P compared to WP. 

The instrumental textural properties and the color characteristics of the pasta were investigated on samples at OCT. The use of black chickpea as ingredients caused a slight but significant (*p* < 0.05) increase of the hardness, while resilience, chewiness and cohesiveness resulted in all cases lower than the semolina control (Table 4). No significant (*p* > 0.05) differences were found in the textural parameters between BC-P and labBC-P.

Significant (*p* < 0.05) differences in the colorimetric coordinates were observed among the pasta samples. Indeed, WP was characterized by the highest *L* (lightness) and *b* (yellow-blue index) values (Table 3), while labBC-P presented the lowest *L* and *b* values, and the highest (although negative) red-green index *a* (Table 4). Consequently, significantly (*p* < 0.05) different *ΔE* values were found (16.79, 22.56 and 20.57, respectively for WP, BC-P and labBC-P).

#### 3.2.2. Nutritional Characterization

The addition of the black chickpea doughs to pasta affected all the biochemical features of the experimental pasta samples. Compared to WP, the use of BC as pasta ingredient led to a significant (*p* < 0.05) increase of the pH, while, as expected, labBC-P was characterized by a pH value ca. 0.5 units lower (Table 4). Compared to WP, significant increases of proteins (9%), fat (10%) and, mostly, dietary fibers (35%) were observed when BC flour, both raw and fermented, was used for pasta production.

Total free amino acids were the highest in labBC-P (602 mg/kg), while significantly (*p* < 0.05) lower concentrations were found in WP (−72%) and BC-P (−29%) (Table 4).

The starch hydrolysis index, determined at OCT, was similar for WP and BC-P, and resulted 18% lower in labBC-P (Table 4). Consequentially, predicted glycemic index (pGI) was 79.8 ± 1.9, 78.1 ± 0.9 and 72.6 ± 1.8 for WP, BC-P and labBC-P, respectively. On the contrary, labBC-P was characterized by the highest IVPD, ca. 62 and 12% higher than WP and BC-P, respectively (Table 4).

#### 3.2.3. Antioxidant Activity

Compared to the antioxidant activity of the black chickpea doughs above discussed, markedly lower values were found in pasta (at OCT), as a consequence of the mixing with semolina, extrusion process and cooking. Nevertheless, the presence of the chickpea flour led to a significantly (*p* < 0.05) higher antioxidant activity in BC-P and labBC-P compared to the semolina control (Figure 3), as determined by using DPPH or ABTS radicals. The inhibition of the peroxidation of the linoleic acid was also significantly (*p* < 0.05) higher in both the fortified pasta samples (Figure 3C). When the labBC was used instead of raw BC dough, significantly (*p <* 0.05) higher DPPH radical scavenging activity and peroxidation inhibition were observed, while similar (*p* > 0.05) ABTS scavenging activity characterized both BC-P and labBC-P (Figure 3).

#### 3.2.4 Sensory Profile

3.2.4 Sensory Profile

A visual inspection of the pasta samples revealed that black chickpea fortification conferred to the products an evident heterogeneity, since small black dots were recognizable on the pasta surface (Figure 4). Pasta texture was also slightly but significantly (*p* < 0.05) affected by the black chickpea addition, since a lack of uniformity and higher stacking (piece-to-piece adhesion) were perceived in BC-P and labBC-P by the assessors, compared to the semolina control (Figure 4). Odor intensity and pungency were judged higher in fortified samples compared to WP, although the greatest differences with the control were found in the flavor attributes (Figure 4). More specifically, BC-P was characterized by intense and pungent legume taste attributes, while labBC-P presented the highest sapidity. No differences (*p* > 0.05) were found in the chewability scores among the three pasta samples.

The overall appreciation of the labBC-P pasta was similar (*p* > 0.05) to WP, while a significantly (*p* < 0.05) lower score was assigned to BC-P.

## 4. Discussion

According to the scientific evidence of the last decades, modern consumers consider food not only as a source of nutrients strictly related to satiety, but also as factors affecting mental health and wellbeing [50]. Indeed, by modulating and keeping a balance within the gut microbiota, food can help prevent cancer, allergies, cardiovascular diseases and inflammatory status [51,52]. 

The fortification of staple foods, such as bread and pasta, with non-conventional vegetable ingredients, has been identified as an effective, sustainable and promising intervention addressing both environmental concerns and nutritional recommendations [9]. Legumes are excellent sources of proteins with high biological value and dietary fibers, also supplying vitamins, minerals, oligosaccharides and phenolic compounds [9]. They represent the best sources of vegetable protein to replace those of animal origin, due to their environmental adaptability, large number of different species and cultivars, global diffusion and positive impact on the soil fertility. In particular, local legume cultivars have been considered important source of biodiversity and therefore increasingly subjected to the exploitation of their technological potential [19,53].

Modern chickpea cultivars, selected for improved agronomic and commercial traits, gradually replaced the black chickpea across the native Mediterranean region, putting it at risk of genetic erosion [24,26]. Based on our knowledge, the effects of LAB fermentation on the black chickpea flour were never investigated before, although sourdough biotechnology (involving LAB) has been largely reported as a suitable tool to enhance nutritional and functional properties of legumes [19,53].

A *Lb. plantarum* strain previously selected for its technological performances [27] was used as a starter for fermentation in this study. Semiliquid doughs (containing 65% of water) were obtained, to mimic the industrial production conditions, in which the use of automatic bioreactors (intended for liquid sourdough propagation) is currently very common. The strain was able to grow to 2 logarithmic cycles reaching a cell density of 10^9^ ufc/g of black chickpea dough. The comparison with a not inoculated dough confirmed its strong competition towards the other microbial groups and its high acidification capacity. 

An intense free amino acids release was observed in fermented doughs, especially when the selected starter was inoculated. The FAA concentration can be considered as an index of the proteolysis degree, already related to the increase of the protein digestibility in foods [54]. Leguminous flours are an excellent supplement for cereal-based food because not only do they increase protein content, but they also improve its biological value, complementing limiting amino acids in cereals (Lys and Trp) with those in legumes (Met and Cys) [55] and in this study, fermentation with the selected starter, allowed a further increase of all the above-mentioned FAA. Besides the nutritional relevance, the high concentration of FAA is associated to the improvement of the sensory profile of the final product [56]. Glu, the main amino acid responsible for the sapidity perception, as previously reported [57], was the most abundant FAA in BC flour and its concentration further increased in LAB-fermented dough. Moreover, the concentration of the functional non-protein amino acid GABA [58] increased in fermented doughs (final concentration higher than 250 mg/kg). Arg and Orn, which were also among the most predominant amino acids in BC, increased and decreased, respectively, during spontaneous fermentation, probably due to the activity of the endogenous microbiota, dominated by presumptive LAB, yeasts, molds and Enterobacteriaceae. As a matter of fact, a similar FAA pattern was observed in wheat dough fermented with a mixture of yeasts, LAB and Enterobacteria [59]. The decrease of some FAA suggests that they were assimilated (used for plastic and functional purposes or catabolized by secondary metabolism patterns) at a higher ratio compared to their release in matrix. Overall, release and accumulation of FAA depends on the activity of microbial proteolytic enzymes and their extracellular translocation when they exceed metabolic requirements [59], thus making not predictable the final balance when a complex microbial consortium is involved in fermentation.

The use of fermentation as pretreatment led to a relevant degradation of the ANF as compared to the corresponding unfermented matrix. Among ANF, α-galactosides of sucrose, such as raffinose, stachyose and verbascose, cause gastrointestinal disorders because they are not degraded in the gastrointestinal tract [60]. Nevertheless, they can be enzymatically hydrolyzed by LAB during fermentation [61], thus increasing product digestibility and reducing digestive discomfort [62]. The degradation of condensed tannins through LAB has already been reported. It involves several enzymatic activities [63], such as tannase, polyphenol oxidase and decarboxylase [18,64,65]. TIA also decreased during fermentation and, as already observed in fermented legumes [21,66], it depends on the specific capability of the LAB strains involved in fermentation [62]. The same was observed for saponins, biologically active glycoside, able to hemolyze red blood cells and to form complexes with nutrients preventing their absorption in small intestine [67], although their anticarcinogenic and blood cholesterol-lowering effects have been also reported [68].

Among non-digestible food components, resistant starch represents a small fraction of starch that is resistant to hydrolysis by digestive enzymes in vitro and in vivo. In legumes, starch digestibility is much lower than that of cereal, which is ascribable to higher content of amylose in the former [69]. In this study, as already reported for other legume-based foods [8,21,70], fermentation with the selected lactic acid bacteria determined a considerable increase of resistant starch content, which is partially due to the acidification caused by the organic acids released during fermentation (Table 2). Besides acidification, other food processing conditions (e.g., thermal treatments) can contribute to the resistant starch increase [69].

Fermentation of the black chickpea with the selected *Lb. plantarum* strain also led to a marked increase of the antioxidant activity as compared to the unfermented or spontaneously fermented doughs. The role of LAB in enhancing antioxidant activity on vegetable matrices was largely reported [71] as the result of the release of antioxidant peptides through proteolysis and, especially, the ability to modify the phenolic profile of the matrix. Polyphenols are often bound to cell wall, glycosylated or in polymeric forms, which affect their bioaccessibility, yet several LAB metabolic activities are involved in their release (e.g., feruloyl esterases, glycosyl hydrolases and tannases) or conversion (e.g., phenolic acid reductases and decarboxylases) into more active forms [71]. 

Aiming at better understanding which compounds were responsible for the antioxidant activity of raw and fermented black chickpea, phenolic compounds were selectively extracted and analyzed by UPLC-PDA-ESI-QTOF. Free and bound phenolic compounds identified in this study were mainly characterized by phenolic acids and glycosides of flavonoids and hydroxybenzoic acid, most of which, previously described in other chickpea cultivars [44,45,47]. As described in Appendix A, other minor metabolites, including soyasaponin isomers, were also found. It was previously reported [26] that Apulian black chickpea accession, compared to other cultivars, have a similar content in total phenolic compounds but are richer in anthocyanins, flavonoids pigments particularly known for their red/purple coloring. Anthocyanins are present in the seed pericarp, explaining the different color of these chickpeas, able to reach a brown hue when, together with carotenoids, are very concentrated [26], however, in this study, they were not identified. Anthocyanins are usually detected using ionization sources in the positive mode [72,73], whereas the ESI system used for this study was in the negative ion mode. It is likely that anthocyanins concentration, which can be highly variable among accessions, ranging from ca. 20 to 160 mg/kg on d.m. [26], was too low to be detected in the negative mode, although possible at more appreciable concentrations. 

The relatively low concentration of phenolic compounds (ca. 90 mg/kg of d.m. between free and bound) was utterly modified by spontaneous and inoculated fermentations, which led to their substantial increment. Whereas few glycosides significantly decreased, the most relevant change observed concerned phloretic and dihydrocaffeic acids, which made up 62% and 82% of the free phenolics in sBC and labBC, respectively. Dihydrocaffeic acid is considered an extremely powerful antioxidant compound, having an antiradical effect higher than that of *α*-tocopherol [74], and also demonstrated ex vivo protecting endothelial cells from oxidative stress in a model in a human-derived one [75]. Usually, phloretic and dihydrocaffeic acids are products of reduction of hydroxycinnamic acids by heterofermentative lactic acid bacteria, which use them as external acceptors of electrons, gaining an additional ATP molecule [76]. In this study, ferulic and isoferulic acids detected in the bound profile of BC decreased after fermentation yet were not found among free compounds of sBC and labBC. It is possible that, as proposed by Perez-Ternero et al. [77], following demethylation and consequent dehydroxylation, caffeic and coumaric were formed, which were then reduced by LAB dehydrogenases to phloretic and dihydrocaffeic [76]. *Lactobacillus* strains and some Enterobacteriaceae, were reported to demethylate and dehydroxylate ferulic acid [78,79]. Moreover, different catabolic pathways leading to their formation have also been proposed for in vitro gastrointestinal digestion. Indeed, Yu et al. [80] observed that quercetin and kaempferol glycosides were subjected to the cleavage of glycoside *c*-linkage resulting in the formation of dihydrocaffeic acid and 3-(4-hydroxyphenyl)-propionic acid (also known as phloretic acid). As a matter of fact, a significant decrease (*p* < 0.05) of quercetin-3,7-*O*-di-glucopyranoside, quercetin-3-*O*-rutinoside-7-*O*-α-L-rhamnopyranoside, and kaempferol 3-*O*-lathyroside-7-*O*-α-L-rhamnopyranoside was observed in free and bound profiles (Table 3). Phloretic and dihydrocaffeic acids, which were the two major colonic catabolites of phenolic compounds, remained in considerable amounts after 24 h of incubation; and Lactobacilli, along with other beneficial bacteria, are known to dominate the gut microbiota [80]. Therefore, it is not unlikely that *L. plantarum* T0A10 (in labBC) and the autochthonous microbiota of the legume (in sBC) produced these compounds in such high amount during black chickpea fermentation; yet more in-depth studies should be performed on the microbial and enzymatic pathways, which led to their formation.

Novel pasta recipes including the replacement of wheat flour with alternative ingredients and the inclusion of pre-fermented ingredients have been recently proposed [15]. Experimental pasta samples were produced using BC and labBC. sBC was not used as an ingredient because the improvements, in terms of nutritional quality, obtained after inoculated fermentation exceeded those obtained in spontaneous fermented chickpea. Moreover, the microbial load of Enterobacteriaceae, molds and yeasts in sBC represents a potential risk for microbiological safety. In order to limit the weakening of the gluten network or the impact on the sensory properties [9], the level of fortification of the black chickpea flour was kept to 15% of the semiliquid dough addition. Lower substitution levels would have not provided an adequate nutritional advantage, whereas higher introduction of non-gluten protein and fibers would have considerably affected the structure of pasta. Despite the moderate semolina replacement, the cooking performances and the textural properties of the fortified pasta were influenced by the inclusion of the chickpea doughs. A decrease in OCT and the increase of cooking loss were indeed observed, regardless the fermentation pretreatment. Nevertheless, fortified pasta was characterized by hardness significantly higher than semolina control, according to previous findings [7,8,9,10].

The addition of the black chickpea doughs to the pasta recipes led to a significant increase in fiber and proteins. In particular, the experimental fortified pasta can be labeled as “source of fiber” and a “source of protein” according to the EC Regulation on nutrition and health claims on food products [81].

Moreover, a relevant increase of IVPD characterized the use of the pre-fermented dough, already associated to the proteolytic process mediated by endogenous and LAB proteinases and peptidases [21,54].

As expected, unfermented black chickpea addition led to the increase of antinutritional compounds concentration in BC-P. However, according to the degradation observed during LAB fermentation, significantly lower concentrations of these compounds were detected when the fermented dough was used as a pasta ingredient.

Pasta containing fermented black chickpea flour had a lower value of HI (and predicted GI) compared to the semolina control. This could be attributed to biological acidification, which is one of the main factors affecting starch hydrolysis rate and HI [42], with the contribution of the dietary fibers and resistant starch higher concentrations [43].

The antioxidant activity in pasta at OCT was estimated with three different in vitro assays, which confirmed the contribution of the black chickpea doughs, especially when it was fermented by the selected LAB. Compared to the literature data, the antioxidant activity was similar to that reported for experimental pasta fortified with fermented quinoa, and other alternative flours, despite the lower semolina replacement level used in this study [7,82].

The sensory profile of the fortified pasta was investigated through a sensory analysis. The color changes observed with the instrumental analysis corresponded to differences recognized by the visual inspection, which also revealed a less homogeneous texture compared to control. The presence of the legume flour was perceived by the panelists both on the taste and smell properties of the cooked pasta, in comparison to the semolina control. Overall, the analysis results described a peculiar but appreciated profile of the fortified samples, especially for the pasta including fermented black chickpea dough, characterized by sapidity and more delicate flavor compared to that containing the unfermented legume. Due to the dilution in cooking water, acidity flavor and smell (commonly related to LAB fermentation) were not perceived in pasta.

## 5. Conclusions

Due to its widespread acceptance pasta has been proposed as a suitable carrier of nutrients, mainly dietary fibers and proteins [1]. Several studies [6,13,34,82] also contemplated the possibility to consider pasta as a potential carrier of antioxidant compounds, like polyphenols, despite the potential susceptibility of their functionality to the thermal treatments [6]. The massive presence of pasta in global diets could counterbalance a relative low antioxidant activity and justify the design and large-scale production of pasta formulations enriched in antioxidant compounds by using alternative ingredients, such as black chickpea flour, also providing high amount of dietary fibers and proteins. This study suggests the use of fermentation by selected LAB as a sustainable and easily applicable biotechnology to overcome the legume flour technological and sensory drawbacks, also positively affecting protein digestibility, glycemic load, antinutritional factors degradation and antioxidant activity.

## Figures and Tables

**Figure 1 foods-10-00182-f001:**
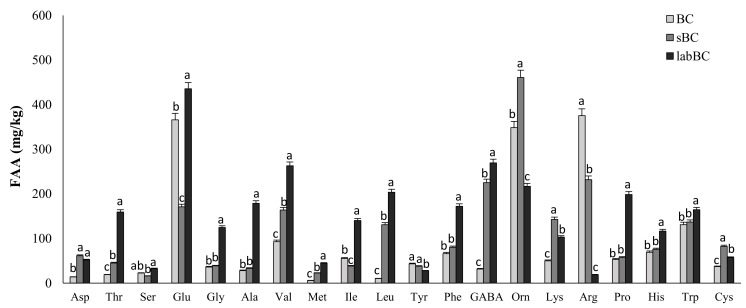
Free amino acids concentration in doughs made with black chickpea flour: BC, not inoculated and not fermented; sBC, spontaneously fermented for 24 h at 30 °C; labBC, inoculated with *L. plantarum* T0A10, and fermented for 24 h at 30 °C. All doughs had DY of 285. ^a–c^ Values with different superscript letters within the same amino acid, differ significantly *(p <* 0.05).

**Figure 2 foods-10-00182-f002:**
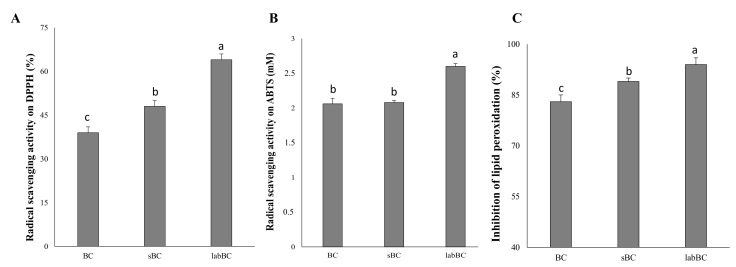
Antioxidant activity of the black chickpea doughs. (**A**) Radical scavenging activity on DPPH radical; (**B**) radical cation ABTS scavenging activity and (**C**) inhibition of the lipid peroxidation (8 days of incubation at room temperature). BC, not inoculated and not fermented; sBC, spontaneously fermented for 24 h at 30 °C; labBC, inoculated with *L. plantarum* T0A10, and fermented for 24 h at 30 °C. ^a–c^ Values with different superscript letters differ significantly (*p* < 0.05).

**Figure 3 foods-10-00182-f003:**
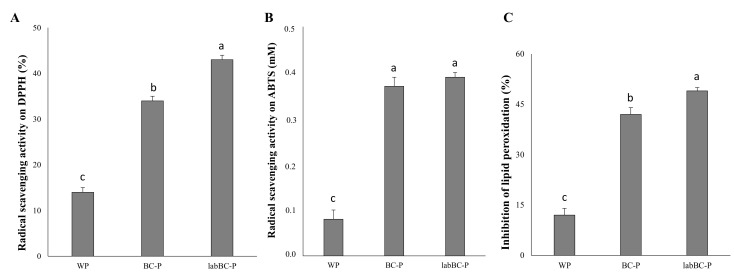
Antioxidant activity of the pasta samples at the OCT: **(A)** radical scavenging activity on DPPH radical; **(B)** radical cation ABTS scavenging activity and **(C)** inhibition of the lipid peroxidation (8 days of incubation at room temperature). BC-P, pasta fortified with 15% (*w/w*) of BC (unfermented black chickpea dough); labBC-P, pasta fortified with 15% (*w/w*) of labBC (fermented black chickpea dough). ^a–c^ Values with different superscript letters differ significantly (*p* < 0.05).

**Figure 4 foods-10-00182-f004:**
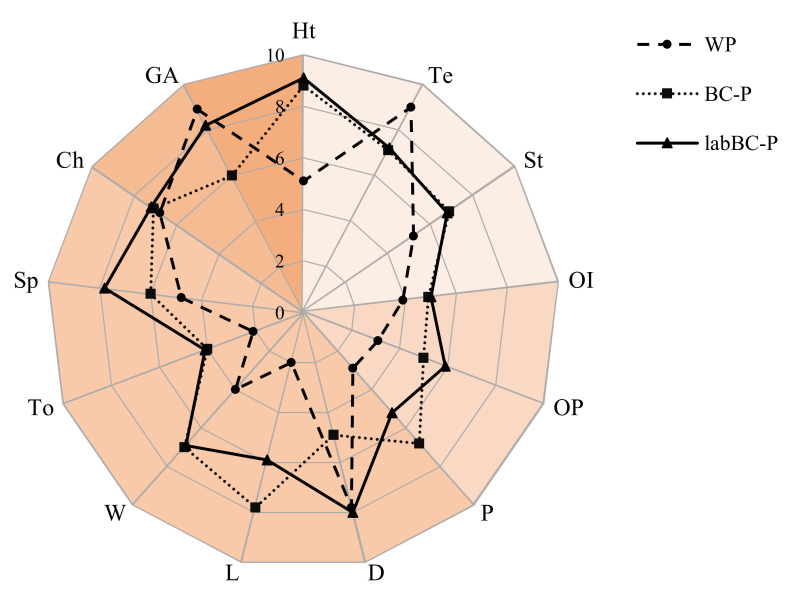
Sensory analysis of the pasta samples at OCT. WP, wheat semolina pasta; BC-P, pasta fortified with 15% (*w/w*) of BC (unfermented black chickpea dough); labBC-P, pasta fortified with 15% (*w/w*) of labBC (fermented black chickpea dough). Visual attributes: color heterogeneity, Ht; texture, Te; stacking, St. Odor attributes: intensity, OI; pungent, OP. Flavor attributes: pungent, P; delicate, D; legume, L; whole, W; toasted, To; sapidity, Sp; Texture attribute: chewability, Ch. General acceptability, GA.

**Table 1 foods-10-00182-t001:** Experimental pasta formulations. WP, wheat semolina pasta used as reference; BC-P, pasta fortified with unfermented black chickpea flour; labBC-P, pasta fortified with fermented black chickpea flour. All formulations had final dough yield (DY) of 130.

	WP	BC-P	labBC-P
Semolina (%)	77	71.7	71.7
Black chickpea flour (%)	-	5.3	-
Fermented black chickpea dough (%) *	-	-	15
Water (%)	23	23	13.3

* Fermented black chickpea dough having DY 285 (64.9% water, 35.1% black chickpea flour) was obtained by fermenting the dough at 30 °C for 24 h with *L. plantarum* T0A10.

**Table 2 foods-10-00182-t002:** Microbiological analysis, biochemical characteristics, in vitro protein digestibility and antinutritional factors concentration in doughs made with black chickpea flour: BC, not inoculated and not fermented; sBC, spontaneously fermented for 24 h at 30 °C; labBC, inoculated with *L. plantarum* T0A10, and fermented for 24 h at 30° C. All doughs had DY of 285.

	BC	sBC	labBC
***Microbiological analysis***			
Lactic acid bacteria (Log cfu/g)	2.54 ± 0.12 ^c^	7.08 ± 0.16 ^b^	9.58 ± 0.11 ^a^
Yeasts (Log cfu/g)	2.02 ± 0.09 ^b^	4.23 ± 0.10 ^a^	2.15 ± 0.13 ^b^
Molds (Log cfu/g)	2.09 ± 0.11 ^b^	3.02 ± 0.09 ^a^	2.33 ± 0.09 ^b^
Enterobacteriaceae (Log cfu/g)	2.27 ± 0.12 ^b^	4.89 ± 0.14 ^a^	2.49 ± 0.11 ^b^
***Biochemical characteristics***			
pH	6.53 ± 0.30 ^a^	4.76 ± 0.23 ^b^	3.85 ± 0.19 ^c^
Total titratable acidity (mL NaOH 0.1 M)	2.0 ± 0.1 ^c^	5.1 ± 0.2 ^b^	10.2 ± 0.5 ^a^
Lactic acid (mmol/kg)	0.4 ± 0.1 ^c^	77.0 ± 2.1 ^b^	123.4 ± 2.5 ^a^
Acetic acid (mmol/kg)	2.14 ± 0.3 ^c^	18.9 ± 0.5 ^a^	13.1 ± 0.8 ^b^
Total free amino acids (mg/kg)	1851 ± 21 ^c^	2249 ± 15 ^b^	2975 ± 18 ^a^
Resistant starch (%)	1.75 ± 0.11 ^c^	2.03 ± 0.12 ^b^	2.81 ± 0.10 ^a^
***Protein digestibility and Antinutritional factors***			
In vitro protein digestibility (%)	80 ± 1^c^	85 ± 1 ^b^	91 ± 2 ^a^
Raffinose (g/kg)	1.27 ± 0.10 ^a^	0.84 ± 0.03 ^b^	0.44 ± 0.02 ^c^
Condensed tannins (g/kg)	0.62 ± 0.02 ^a^	0.36 ± 0.03 ^b^	0.34 ± 0.02 ^b^
TIA* (U)	0.66 ± 0.05 ^a^	0.59 ± 0.05 ^b^	0.52 ± 0.04 ^b^
Total saponins (g/kg)	0.68 ± 0.02 ^a^	0.45 ± 0.03 ^b^	0.29 ± 0.05 ^c^

* Trypsin inhibitor activity was expressed as trypsin inhibitor units/mg sample. The data are the means of three independent experiments ± standard deviations (*n* = 3). ^a–c^ Values in the same row with different superscript letters differ significantly (*p* < 0.05).

**Table 3 foods-10-00182-t003:** Concentration, expressed as mg/kg (d.m.), of free and bound phenolic compounds in black chickpea doughs by UPLC-PDA-ESI-QTOF. BC, dough made with black chickpea, not inoculated and not fermented; sBC, spontaneously fermented dough made with black chickpea for 24 h at 30 °C; labBC, dough made with black chickpea, inoculated with *L. plantarum* T0A10, and fermented for 24 h at 30° C. All doughs had a DY of 285.

	BC	sBC	labBC
***Free phenolic compounds***			
Protocatechuic acid	3.72 ± 0.19 ^c^	5.51 ± 0.28 ^b^	10.57 ± 0.53 ^a^
Dihydroxybenzoic acid hexoside	2.65 ± 0.13	nd	nd
Dihydrocaffeic acid	nd	5.98 ± 0.30 ^b^	93.35 ± 4.67 ^a^
Hydroxybenzoic acid hexoside pentoside	5.70 ± 0.28 ^c^	8.35 ± 0.37 ^b^	9.87 ± 0.49 ^a^
Dihydroxybenzoic acid hexoside-pentoside I	0.36 ± 0.02 ^b^	0.66 ± 0.03 ^a^	0.77 ± 0.04 ^a^
Dihydroxybenzoic acid hexoside-pentoside II	0.99 ± 0.05 ^b^	1.46 ± 0.07 ^a^	1.67 ± 0.08 ^a^
Phloretic acid	nd	48.24 ± 2.41 ^b^	102.00 ± 5.10 ^a^
Hydroxybenzoic acid hexoside-pentoside dehydrodimer	2.56 ± 0.13 ^c^	3.87 ± 0.18 ^b^	4.49 ± 0.22 ^a^
Hydroxybenzoic acid hexoside-pentoside trimer	0.36 ± 0.02 ^c^	0.59 ± 0.03 ^b^	0.70 ± 0.03 ^a^
Hydroxibenzoic acid derivative	0.56 ± 0.03 ^b^	1.44 ± 0.07 ^a^	1.52 ± 0.08 ^a^
Myricetin derivative	1.06 ± 0.05 ^c^	1.40 ± 0.07 ^b^	1.71 ± 0.09 ^a^
Quercetin-3,7-*O*-di-glucopyranoside	0.76 ± 0.04 ^a^	0.30 ± 0.04 ^b^	0.29 ± 0.05 ^b^
Quercetin-3-*O*-b-D-xylopyranosyl-(1/2)-rutinoside	1.44 ± 0.07 ^c^	1.66 ± 0.09 ^b^	2.03 ± 0.10 ^a^
Myricetin-*O*-methylether hexoside-deoxyhexoside-pentoside	3.63 ± 0.18 ^c^	4.87 ± 0.24 ^b^	5.25 ± 0.26 ^a^
Kaempferol 3-*O*-lathyroside-7-*O*-α-L-rhamnopyranoside	1.43 ± 0.07 ^c^	1.87 ± 0.09 ^a^	2.01 ± 0.10 ^a^
Quercetin-3-*O*-rutinoside-7-*O*-α-L-rhamnopyranoside	1.46 ± 0.05 ^a^	1.03 ± 0.07 ^b^	0.98 ± 0.07 ^b^
**Total**	**26.67 ± 1.09 ^c^**	**87.22 ± 2.97 ^b^**	**237.02 ± 10.95 ^a^**
***Bound phenolic compounds***			
Gallic acid	9.66 ± 0.48 ^a^	7.29 ± 0.36 ^b^	8.48 ± 0.42 ^a^
Protocatechuic acid	29.31 ± 1.47 ^a^	18.88 ± 0.94 ^c^	25.32 ± 1.27 ^b^
Dihydroxybenzoic acid hexoside	2.24 ± 0.11 ^a^	0.34 ± 0.02 ^b^	0.32 ± 0.02 ^b^
Hydroxybenzoic acid hexoside-pentoside	1.34 ± 0.07 ^a^	0.76 ± 0.04 ^c^	1.12 ± 0.06 ^b^
Morin	1.43 ± 0.07	1.44 ± 0.07	1.62 ± 0.10
Dihydroxybenzoic acid hexoside-pentoside I	0.41 ± 0.02 ^a^	0.34 ± 0.01 ^b^	0.33 ± 0.01 ^b^
Ferulic acid	7.63 ± 0.38 ^a^	4.35 ± 0.22 ^b^	4.47 ± 0.22 ^b^
Isoferulic acid	19.73 ± 0.99 ^a^	13.96 ± 0.70 ^c^	15.97 ± 0.80 ^b^
Myricetin-*O*-methyl ether hexoside-deoxyhexoside-pentoside	0.71 ± 0.04 ^a^	0.10 ± 0.01 ^b^	0.16 ± 0.01 ^b^
Kaempferol 3-*O*-lathyroside-7-*O*-α-L-rhamnopyranoside	0.62 ± 0.03 ^a^	0.19 ± 0.02 ^b^	0.19 ± 0.02 ^b^
**Total**	**73.09 ± 2.94 ^a^**	**47.54 ± 2.12 ^c^**	**57.88 ± 3.05 ^b^**

The data are the means of three independent experiments ± standard deviations (*n* = 3). ^a–c^ Values in the same row with different superscript letters differ significantly (*p* < 0.05). nd: not detected.

**Table 4 foods-10-00182-t004:** Chemical, nutritional, and technological features of pasta. WP, pasta made with wheat semolina; BC-P, pasta fortified with 15% (*w/w*) of BC (unfermented black chickpea dough); labBC-P, pasta fortified with 15% (*w/w*) of labBC (fermented black chickpea dough).

	WP	BC-P	labBC-P
***Chemical and nutritional characteristics***			
pH	6.47 ± 0.02 ^b^	6.91 ± 0.01 ^a^	5.97 ± 0.02 ^c^
Moisture (%)	8.9 ± 0.2	9.1 ± 0.2	9.2 ± 0.1
Proteins (% of d.m.)	12.6 ± 0.2 ^b^	13.7 ± 0.2 ^a^	13.6 ± 0.2 ^a^
Fat (% of d.m.)	2.0 ± 0.1 ^b^	2.2 ± 0.2 ^a^	2.2 ± 0.2 ^a^
Available carbohydrates (% of d.m.)	79.0 ± 1.2 ^a^	77.1 ± 1.5 ^a b^	75.9 ± 0.8 ^b^
Dietary fibers (% of d.m.)	4.0 ± 0.1 ^b^	5.2 ± 0.2 ^a^	5.4 ± 0.1 ^a^
Ash (% of d.m.)	0.84 ± 0.04 ^b^	0.95 ± 0.02 ^a^	1.01 ± 0.02 ^a^
Total Free Amino Acids (mg/kg)	152 ± 4 ^c^	429 ± 6 ^b^	602 ± 11 ^a^
Hydrolysis Index (%) *	73 ± 2 ^a^	70 ± 1 ^a^	60 ± 2 ^b^
In Vitro Protein Digestibility (%) *	45 ± 1 ^c^	65 ± 2 ^b^	73 ± 2 ^a^
***Technological and textural*** *** *characteristics***			
Optimal Cooking Time (min)	9.7 ± 0.2 ^a^	6.9 ± 0.2 ^b^	6.3 ± 0.1 ^c^
Water absorption (g/100 g)	128.6 ± 1.2 ^a^	113.1 ± 1.5 ^b^	115.3 ±1.5 ^b^
Cooking loss (g/100 g)	4.70 ± 0.12 ^b^	5.73 ± 0.14 ^a^	5.80 ± 0.08 ^a^
Hardness (N)	4.31 ± 0.11 ^b^	4.89 ± 0.09 ^a^	4.80 ± 0.05 ^a^
Resilience	0.68 ± 0.04 ^a^	0.58 ± 0.05 ^b^	0.58 ± 0.02 ^b^
Chewiness (N)	2.86 ± 0.05 ^a^	2.56 ± 0.03 ^b^	2.53 ± 0.03 ^b^
Cohesiveness	0.59 ± 0.02 ^a^	0.44 ± 0.02 ^b^	0.49 ± 0.03 ^b^
***Color analysis*** *****			
L	66.42 ± 0.20 ^a^	44.18 ± 0.18 ^b^	46.57 ± 0.21 ^b^
a	−6.95 ± 0.05 ^b^	−7.14 ± 0.08 ^c^	−1.67 ± 0.03 ^a^
b	20.61 ± 0.03 ^a^	14.16 ± 0.05 ^b^	12.78 ± 0.04 ^c^
ΔE	16.79 ^c^	22.56 ^a^	20.57 ^b^

* Hydrolysis index, in vitro protein digestibility, textural profile analysis and color analysis refer to pasta at the optimal cooking time (OCT). Data are the means of three independent experiments ± standard deviations (*n* = 3). ^a–c^ Values in the same row with different superscript letters differ significantly (*p* < 0.05).

## Data Availability

The data presented in this study are available upon request from the corresponding author.

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
