# Peer review of "Nutritional and Functional Advantages of the Use of Fermented Black Chickpea Flour for Semolina-Pasta Fortification"

_foods, 2021, doi:10.3390/foods10010182_

Round 1
Reviewer 1 Report
In the manuscript, the authors describe an interesting approach to improving the nutritional quality of semolina pasta by adding black chickpea flour. The supplementation with a Mediterranean black chickpea flour increased some nutrition value of semolina pasta (protein, total free amino acids, dietary fibers), but on the other hand, the addition of the mentioned flour was also the source of some antinutritional factors (raffinose, condensed tannins, trypsin inhibitors and saponins). The authors made an attempt to reduce these unfavorable components by fermentation of semi-liquid dough of a Mediterranean black chickpea with Lactiplantibacillus. As a result, the positive effect of fermentation process on the final quality of the supplemented product was shown. The results has been well documented by numerous outcomes of the analyzes (physical and chemical indicators and sensory analysis). The paper is quite fairly written and organized. Below, there are some comments and suggestions:
Line 41: If 14,5 million tons of pasta is produced annually and the revenues are in approximation 99 mln, one ton of pasta costs 6,83 USD, what is unlikely.
l. 43: Sentence: “…..nutritional features, such as on the antinutritional (ANF) compounds were studied. “ is not clear. It seems that the mentioned sentence needs a correction.
l. 103: The authors do not explain how the late exponential phase of growth mentioned in the sentence: “Prior to be inoculated, L. plantarum T0A10 was cultivated until the late exponential phase of growth was reached (approx. 12 h)” was determined. It would be advisable to provide a graph with the CFU changes over time.
l. 307: Could the authors try to explain why the amount of some amino acids increased and others decreased.
l. 355: In the title of subsection 3.1.4. “Identification of free and bound compounds” the authors should consider adding the word “phenolic” before compounds.
l.326: Table 2 - microbiological analysis section: the values and their standard deviations have different range of accuracy.
Author Response
In the manuscript, the authors describe an interesting approach to improving the nutritional quality of semolina pasta by adding black chickpea flour. The supplementation with a Mediterranean black chickpea flour increased some nutrition value of semolina pasta (protein, total free amino acids, dietary fibers), but on the other hand, the addition of the mentioned flour was also the source of some antinutritional factors (raffinose, condensed tannins, trypsin inhibitors and saponins). The authors made an attempt to reduce these unfavorable components by fermentation of semi-liquid dough of a Mediterranean black chickpea with Lactiplantibacillus. As a result, the positive effect of fermentation process on the final quality of the supplemented product was shown. The results has been well documented by numerous outcomes of the analyzes (physical and chemical indicators and sensory analysis). The paper is quite fairly written and organized. Below, there are some comments and suggestions:
Line 41: If 14,5 million tons of pasta is produced annually and the revenues are in approximation 99 mln, one ton of pasta costs 6,83 USD, what is unlikely.
The authors thank the reviewer for the comment. As a matter of fact, there was a mistake. The correct revenue is 99 billion USD, which means one ton of pasta costs on average ca. 7,000 USD. The right revenue data was revised in the text (line 41).
- 43: Sentence: “…..nutritional features, such as on the antinutritional (ANF) compounds were studied. “ is not clear. It seems that the mentioned sentence needs a correction.
The sentence has been rephrased as suggested (lines 83-84).
- 103: The authors do not explain how the late exponential phase of growth mentioned in the sentence: “Prior to be inoculated, L. plantarum T0A10 was cultivated until the late exponential phase of growth was reached (approx. 12 h)” was determined. It would be advisable to provide a graph with the CFU changes over time.
Parameters for the kinetics of growth of the strain have been added in the revised material and methods section (Lines 107-112)
- 307: Could the authors try to explain why the amount of some amino acids increased and others decreased.
Besides being fermentation metabolites, amino acids are important in the metabolism of sourdough microorganisms, they are among the nutritional requirements necessary to promote growth and proteolytic activities. Therefore, their reduction suggests that they were assimilated (used for plastic and functional purposes or converted into derivatives) at a higher extent than they were released by proteolysis. This concept was explained in the text together with the appropriate references (lines 547-555).
- 355: In the title of subsection 3.1.4. “Identification of free and bound compounds” the authors should consider adding the word “phenolic” before compounds.
Revised as suggested
l.326: Table 2 - microbiological analysis section: the values and their standard deviations have different range of accuracy.
The table was revised according to reviewer’s suggestion.
Reviewer 2 Report
The manuscript “Nutritional and Functional Advantages of the Use of Fermented Black Chickpea Flour for Semolina-Pasta Fortification” reports the outcomes of a comprehensive study on pasta made with unfermented and LAB-fermented flour of an Italian ecotype of black chickpea.
The manuscript is very interesting and makes a contribution to the valorization of local crops, as well as to the improvement of traditional foods, such as pasta, which can thus become a vehicle of healthy components and contribute to the WHO Target of making diets healthier.
Please, find below some suggestions:
Line 2-4: please, delete the full-stop from the title.
Table 1: what does “a” in apex refer to?
Line 209-229: In lines 98-108, you stated that black chickpea (BC) flour was
- Inoculated with LAB (labBC)
- Fermented spontaneously (sBC), and
- Non inoculated (ct-BC)
Now, in lines 209-229, you write that you produced pasta with
- labBC, so you got labBC-P
- non-ioculated/unfermented BC, so you got BC-P
- wheat semolina, so you got WP
now, my question is: what about pasta with the fermented spontaneously black chickpea flour? My suggestion is to add a sentence, where you state why the spontaneous fermented flour was not used to produce pasta.
In addition, my suggestion is to create a certain correspondence between the used acronyms for dough and pasta. For instance, you used ct-BC (standing for control black chickpea, I guess, for flour), but then in the pasta making section you rightly state that your control is wheat semolina pasta. So, my suggestion is to delete “ct-“ and leave BC, or use uBC so that confusion does not arise.
Line 212-214: I would suggest moving these lines after presenting BC pasta, as WS will be the ingredient used in the control pasta and is presented afterwards.
Line 244: you already defined “dry matter”
Line 250: please, define IVDP
Line 300: you state that “very high concentration of free amino acids (FAA) was found in chickpea dough” which one? Above, you discussed results for sBC, labBC and ct-BC.. Is it a general statement referring to ALL doughs? If, so, please, add the s or specify which one you are referring to.
Line 301: please, replace “significantly” with “significant”
Line 303: BC: which one? Did you mean ct-BC? Please, make consistent with the previous acronyms. Also based on the previous comments, I think BC is better than ct-BC, however it is important that everything is consistent all through the text, otherwise confusion arises. One more suggestion is to use uBC to help the reader, as suggested above.
Figure 1: in the legend, you wrote BC, please, make consistent. In addition, please, use the dot as decimal separator and not the comma. Please, add in the caption what A), B) and C) are. Please, make the layout (e.g., size font) of the caption consistent to the Journal guidelines.
Line 320: please, replace “significantly” with “significant”
Line 334: the title reports “Total phenols”, but the text does not make any reference to this parameter. Please, amend the text or add content to the paragraph.
Line 423: please amend “valus” in “values”
Line 439: please, replace “significantly” with “significant”
Line 457: “black chickpea”: do you mean black chickpea doughs? Please, specify.
Figure 2: in the legend, you wrote BC, please, make consistent. Please, use the dot as decimal separator and not the comma. Please, add in the caption what A), B) and C) are. Please, make the layout (e.g., size font) consistent to the Journal guidelines.
Line 488-494: Please, make the layout (e.g., size font) of the caption consistent to the Journal guidelines.
Line 497-500: English language revision required.
Line 567: please, make “soyasaponins” in two words.
Line 562-600: this paragraph is too long, please, try to split it in shorter paragraphs according to the different topics treated.
Author Response
The manuscript “Nutritional and Functional Advantages of the Use of Fermented Black Chickpea Flour for Semolina-Pasta Fortification” reports the outcomes of a comprehensive study on pasta made with unfermented and LAB-fermented flour of an Italian ecotype of black chickpea.
The manuscript is very interesting and makes a contribution to the valorization of local crops, as well as to the improvement of traditional foods, such as pasta, which can thus become a vehicle of healthy components and contribute to the WHO Target of making diets healthier.
Please, find below some suggestions:
Line 2-4: please, delete the full-stop from the title.
Revised as requested.
Table 1: what does “a” in apex refer to?
It was an oversight. The asterisk already specifies how fermented chickpea dough was made.
Line 209-229: In lines 98-108, you stated that black chickpea (BC) flour was
- Inoculated with LAB (labBC)
- Fermented spontaneously (sBC), and
- Non inoculated (ct-BC)
Now, in lines 209-229, you write that you produced pasta with
- labBC, so you got labBC-P
- non-ioculated/unfermented BC, so you got BC-P
- wheat semolina, so you got WP
now, my question is: what about pasta with the fermented spontaneously black chickpea flour? My suggestion is to add a sentence, where you state why the spontaneous fermented flour was not used to produce pasta.
Spontaneous fermented chickpea was not used to produce pasta because the improvements obtained after fermentation, in terms of nutritional quality (reduction of antinutritional factors, increase of free amino acids, protein digestibility and bioavailability of phenolic compounds etc.), were markedly higher when fermentation was carried out with the selected starters compared to spontaneously fermented dough. Moreover, as microbiological analysis showed, the microbial load of “undesired” microorganisms (Enterobacteriaceae, molds and yeasts) increased after incubation without starters. This would have posed a risk in terms of microbiological safety of the pasta potentially produced, but most of all, the variability of the raw material after spontaneously fermentation would have not guaranteed that level of standardization that inoculated fermentation ensures.
This choice has been better justified in the discussion section (lines 631-635)
In addition, my suggestion is to create a certain correspondence between the used acronyms for dough and pasta. For instance, you used ct-BC (standing for control black chickpea, I guess, for flour), but then in the pasta making section you rightly state that your control is wheat semolina pasta. So, my suggestion is to delete “ct-“ and leave BC, or use uBC so that confusion does not arise.
The reviewer suggestion is very much appreciated. Acronyms for doughs have been changed as follows in the text, figures and tables: BC, non-fermented black chickpea dough; sBC, spontaneously fermented black chickpea dough; labBC, fermented black chickpea dough with selected lactic acid bacteria.
Line 212-214: I would suggest moving these lines after presenting BC pasta, as WS will be the ingredient used in the control pasta and is presented afterwards.
The sentence has been revised as suggested (lines 225-227).
Line 244: you already defined “dry matter”
Yes, the sentence has been revised.
Line 250: please, define IVDP
It was an oversight. It is IVPD (In Vitro Protein Digestibility). The abbreviation has been corrected in the text.
Line 300: you state that “very high concentration of free amino acids (FAA) was found in chickpea dough” which one? Above, you discussed results for sBC, labBC and ct-BC.. Is it a general statement referring to ALL doughs? If, so, please, add the s or specify which one you are referring to.
Yes, the statement referred to all doughs. The “s” was added.
Line 301: please, replace “significantly” with “significant”
Done.
Line 303: BC: which one? Did you mean ct-BC? Please, make consistent with the previous acronyms. Also based on the previous comments, I think BC is better than ct-BC, however it is important that everything is consistent all through the text, otherwise confusion arises. One more suggestion is to use uBC to help the reader, as suggested above.
Acronyms for doughs are now consistent in the text, figures and tables.
Figure 1: in the legend, you wrote BC, please, make consistent. In addition, please, use the dot as decimal separator and not the comma. Please, add in the caption what A), B) and C) are. Please, make the layout (e.g., size font) of the caption consistent to the Journal guidelines.
Revised as requested.
Line 320: please, replace “significantly” with “significant”
Ok.
Line 334: the title reports “Total phenols”, but the text does not make any reference to this parameter. Please, amend the text or add content to the paragraph.
The title has been changed. Phenolic compounds have been described in the following paragraph (366-401)
Line 423: please amend “valus” in “values”
Done.
Line 439: please, replace “significantly” with “significant”
Ok.
Line 457: “black chickpea”: do you mean black chickpea doughs? Please, specify.
Yes, the authors referred to black chickpea doughs. The sentence has been revised.
Figure 2: in the legend, you wrote BC, please, make consistent. Please, use the dot as decimal separator and not the comma. Please, add in the caption what A), B) and C) are. Please, make the layout (e.g., size font) consistent to the Journal guidelines.
Revised as requested.
Line 488-494: Please, make the layout (e.g., size font) of the caption consistent to the Journal guidelines.
Revised as requested.
Line 497-500: English language revision required.
The sentence has been corrected.
Line 567: please, make “soyasaponins” in two words.
Among the compounds identified (Table S3) several isomers of the compound “Soyasaponin” were identified. Soyasaponin is the name of the compound known with its IUPAC nomenclature “(2S,3S,4S,5R,6R)-6-[[(3S,4S,4aR,6aR,6bS,8aR,9R,12aS,14aR,14bR)-9-hydroxy-4-(hydroxymethyl)-4,6a,6b,8a,11,11,14b-heptamethyl-1,2,3,4a,5,6,7,8,9,10,12,12a,14,14a-tetradecahydropicen-3-yl]oxy]-5-[(2S,3R,4S,5R,6R)-4,5-dihydroxy-6-(hydroxymethyl)-3-[(2S,3R,4R,5R,6S)-3,4,5-trihydroxy-6-methyloxan-2-yl]oxyoxan-2-yl]oxy-3,4-dihydroxyoxane-2-carboxylic acid”. More information about Soyasaponin can be found on PubChem (https://pubchem.ncbi.nlm.nih.gov/compound/Soyasaponin-I#section=IUPAC-Name ).
The concept has been better clarified in the text (lines 593-594).
Line 562-600: this paragraph is too long, please, try to split it in shorter paragraphs according to the different topics treated.
Revised as requested.
Reviewer 3 Report
The paper titled “Nutritional and Functional Advantages of the Use of Fermented Black Chickpea Flour for Semolina-Pasta Fortification” deals within the scope of the Foods Journal, by investigating an interesting topic of research. This is an extensive study (maybe even too extensive) and certainly can contribute to the field of fortified pasta production.
However, some small improvements can be done. Please find below some remarks to help the revision of the manuscript.
Corrections to be made:
Line 21: “black chickpea flour” instead “black chickpea”
Line 66: Authors should include one section of the literature overview on the impact of fermentation process on enhancing nutritional quality of flour (pasta).
Line 74: Please, elaborate the statement “They have high-quality protein…”. Is it biological value?
Line 224: What was the total drying time?
Line 225 (Table 1): Please correct the “chickpea four” to “chickpea flour” in the Table.
Line 232: There are no hydration test results in the manuscript?
Line 250: Is it IVDP or IVPD?
Line 255: “The predicted glycemic index (pGI)”
Line 260: I assume the paste was drained with a colander?
Line 310: It would be nice to add some reference regarding the influence of fermentation on the increase of RS content.
Line 422: Please, add OCT units – minutes.
Line 448 (Table 4): Unit for Chewiness is “N”.
Line 537: It would be nice to add some discussion and references regarding the influence of fermentation on the increase of RS content.
Author Response
The paper titled “Nutritional and Functional Advantages of the Use of Fermented Black Chickpea Flour for Semolina-Pasta Fortification” deals within the scope of the Foods Journal, by investigating an interesting topic of research. This is an extensive study (maybe even too extensive) and certainly can contribute to the field of fortified pasta production.
However, some small improvements can be done. Please find below some remarks to help the revision of the manuscript.
The authors thank the reviewer for the comments. Changes to the manuscript according to reviewer’s suggestions have been made.
Corrections to be made:
Line 21: “black chickpea flour” instead “black chickpea”
Ok.
Line 66: Authors should include one section of the literature overview on the impact of fermentation process on enhancing nutritional quality of flour (pasta).
A short paragraph, citing one of the most recent and thorough review on the subject has been added as requested (Lines 66-69), while a more extensive discussion concerning each specific results obtained is included in the final section of the article. References and comments to the literature studies concerning the effects of fermentation on the nutritional quality of vegetable matrices and their use as ingredient in food products were included.
Line 74: Please, elaborate the statement “They have high-quality protein…”. Is it biological value?
The statement has been rephrased. In the literature there are no studies related to the biological value or nutritional index of black chickpea.
Line 224: What was the total drying time?
Pasta was dried according to the cycle in Table S1. More specifically, the cycle was composed of 28 steps of ventilation and recovery having specific temperature and humidity for a total of ca. 8 hours. This concept has been clarified in the text (line 233).
Line 225 (Table 1): Please correct the “chickpea four” to “chickpea flour” in the Table.
Done
Line 232: There are no hydration test results in the manuscript?
Yes, there was a mistake. The results of the hydration test have been added to the revised text (lines 432-434).
Line 250: Is it IVDP or IVPD?
It is IVPD (In Vitro Protein Digestibility). The abbreviation was corrected in the text.
Line 255: “The predicted glycemic index (pGI)”
Revised
Line 260: I assume the paste was drained with a colander?
Of course, pasta was drained. Details have been added in the text (line 268).
Line 310: It would be nice to add some reference regarding the influence of fermentation on the increase of RS content.
A sentence explaining the influence of fermentation on resistant starch was added in the discussion section (line 573-580).
Line 422: Please, add OCT units – minutes.
Done.
Line 448 (Table 4): Unit for Chewiness is “N”.
Yes, unit for chewiness is Newton. “Ch” is the abbreviation the authors chose for this parameter. A correction has been made.
Line 537: It would be nice to add some discussion and references regarding the influence of fermentation on the increase of RS content.
A short paragraph explaining the influence of fermentation on resistant starch was added (line 573-580)
Round 2
Reviewer 2 Report
Dear Authors,
thank you for amending the manuscript according to suggestions.
Author Response
The authors thank the reviewer.